# Digital health in fragile states in the Middle East and North Africa (MENA) region: A scoping review of the literature

Fadi El-Jardali[1,2,3]*, Lama Bou-Karroum[1,2], Mathilda Jabbour[1,2], Karen Bou-Karroum[1], Andrew Aoun[1], Sabine Salameh[1], Patricia Mecheal[4], Chaitali Sinha[5]

1 Department of Health Management and Policy, Faculty of Health Sciences, American University of Beirut, Beirut, Lebanon, 2 Knowledge to Policy Center, Faculty of Health Sciences, American University of Beirut, Beirut, Lebanon, 3 Department of Health Research Methods, Evidence, and Impact, McMaster University, Hamilton, Ontario, Canada, 4 HealthEnabled, Washington, DC, United States of America, 5 International Development Research Centre, Ottawa, Ontario, Canada

* fe08@aub.edu.lb

## Abstract

### Introduction

Conflict, fragility and political violence, that are taking place in many countries in the Middle East and North Africa (MENA) region have devastating effects on health. Digital health technologies can contribute to enhancing the quality, accessibility and availability of health care services in fragile and conflict-affected states of the MENA region. To inform future research, investments and policy processes, this scoping review aims to map out the evidence on digital health in fragile states in the MENA region.

### Method

We conducted a scoping review following the Joanna Briggs Institute (JBI) guidelines. We conducted descriptive analysis of the general characteristics of the included papers and thematic analysis of the key findings of included studies categorized by targeted primary users of different digital health intervention.

### Results

Out of the 10,724 articles identified, we included 93 studies. The included studies mainly focused on digital health interventions targeting healthcare providers, clients and data services, while few studies focused on health systems or organizations managers. Most of the included studies were observational studies (49%). We identified no systematic reviews. Most of the studies were conducted in Lebanon (32%) followed by Afghanistan (13%) and Palestine (12%). The first authors were mainly affiliated with institutions from countries outside the MENA region (57%), mainly United Kingdom and United States. Digital health interventions provided a platform for training, supervision, and consultation for health care providers, continuing education for medical students, and disease self-management. The

**Data Availability Statement:** All relevant data are within the manuscript and its Supporting Information files.

**Funding:** This work is supported by the International Development Research Centre in Canada. The funding has ended. The funders had no role in study design, data collection and analysis, decision to publish, or preparation of the manuscript.

**Competing interests:** The authors have declared that no competing interests exist.

review also highlighted some implementation considerations for the adoption of digital health such as computer literacy, weak technological infrastructure, and privacy concerns.

## Conclusion

This review showed that digital health technologies can provide promising solutions in addressing health needs in fragile and conflict-affected states. However, rigorous evaluation of digital technologies in fragile settings and humanitarian crises are needed to inform their design and deployment.

## Introduction

Conflict, fragility and political violence, that are taking place in many countries in the Middle East and North Africa (MENA) region have devastating effects on health and present major barriers to the achievement of the Sustainable Development Goals (SDGs). More than 16 million are forcibly displaced in the MENA region [1] with around 6 million Syrian refugees distributed across Turkey, Lebanon, Iraq, Jordan, and Egypt [2]. High rates of non-communicable diseases (NCDs) and mental health disorders were observed among Syrians refugees and internally displaced [3, 4]. A recent systematic reviews assessing the health needs of displaced Syrians identified mental health and women's health as the greatest health needs in the region [5]. The collapse of the public health infrastructure and sanitation services in war-affected countries resulted in the spread of infectious diseases such as the cholera outbreak in Yemen [6, 7]. Conflict also negatively affected the health care system and limited the accessibility and availability of healthcare professionals and services leading to more morbidity and mortality among civilians [6].

In such challenging circumstances, where health-related infrastructure and systems are stressed or damaged, digital health technologies such as telehealth, mobile health and wireless health devices can be a key component within solutions to enhance quality, accessibility and availability of health care services [6, 8]. Innovations using digital health modalities can also assist in collecting and managing accurate data and enhancing inter-organizational communication efforts. They can aid in the transformation of conflict-affected areas and fragile states to meet the global health goals [6]. However, the context and cultural sensitivity in the MENA region and the vulnerability of women and children in conflict and fragile settings require heightened considerations to privacy and security issues. The implementation of digital health intervention requires ensuring the ethical and legal considerations are in place to in order to protect women and children's health data [9].

The COVID-19 pandemic highlighted the importance of digital health in effectively responding and containing this public health crisis [10, 11]. In low and middle-income countries, digital health, in particular telemedicine, was found to reduce transmission risk and improve access to healthcare during COVID-19 [12]. Despite the reported benefits of locally relevant and appropriately designed digital technologies, they are still underutilized in fragile states in the MENA region. Digital health research is also still in its infancy in this region with evident disparities in comparison to the global context [13]. To inform future research, investments and policy, the objective of this scoping review is to map out the evidence on digital health in fragile states in the MENA region. To the best of our knowledge this is the first scoping review to focus on the region. Previous reviews focused on the role of digital health in

humanitarian settings [14] or in conflict-affected population [6] in general with no specific focus on the MENA region.

## Methods

### Protocol and registration

We did not register the scoping review protocol as PROSPERO do not accept scoping review protocols for registration. The protocol is available upon request.

### Scoping review method

We conducted a scoping review, which is typically used to present *"a broad overview of the evidence pertaining to a topic, irrespective of study quality, to examine areas that are emerging, to clarify key concepts and to identify gaps"*. We followed Joanna Briggs Institute (JBI) guidelines for conducting scoping reviews [15] and the PRISMA Extension for Scoping Reviews (PRISMA-ScR) for reporting scoping reviews [16].

### Definitions

We provide below the definitions of terms used in this scoping review:

Fragile states: We defined fragility as '*a combination of exposure to risk and insufficient coping capacity of the state, systems and/or communities to manage, absorb or mitigate those risks*'. The World Bank Group identifies annually a list of fragile states including "countries with institutional fragility" and "countries affected by violent conflict".

Digital health is defined as an "*the systematic application of information and communications technologies, computer science and data to support informed decision-making by individuals, the health workforce and health systems, to strengthen resilience to disease and improve health and wellness for all*" [17]. It includes but not limited to telemedicine, e-health, mhealth, electronic medical records, and wearable devices. We used the WHO classification of digital health interventions to understand the different interventions in the literature. The digital health interventions are organized into the following categories based on the targeted primary users [18]:

○ Clients: Clients are members of the public who are potential or current users of health services, including health promotion activities. Caregivers of clients receiving health services are also included in this group.

○ Healthcare providers: Healthcare providers are members of the health workforce who deliver health services.

○ Health system managers: Health system managers are involved in the administration and oversight of public health systems. Interventions within this category reflect managerial functions related to supply chain management, health financing, human resource management.

○ Data services: This consists of crosscutting functionality to support a wide range of activities related to data collection, management, use, and exchange.

### Eligibility criteria

- Setting of interest: We included studies on fragile states from the MENA region. We considered fragility caused by conflicts as well as institutional fragility. As per the World Bank 2019 classification of fragile and conflict-affected situations, the fragile states in the MENA region are: Afghanistan, Djibouti, Iraq, Lebanon, Libya, Somalia, Sudan, Syria, West bank and Gaza, and Yemen [19]. We restricted our eligibility to articles published after the year 2010 as this region experienced significant changes after this year mainly the Arab Spring which began in 2010 and sparked the protests, uprisings and conflict throughout the region.

- Intervention of interest: We included studies discussing digital health interventions in health services, health systems and public health. We excluded digital health interventions in clinical settings.

- Design of interest: We included any study design except for protocols, news articles, letters, opinion pieces, abstracts and posters.

### Literature search

We searched the following electronic databases: Ovid Medline, PubMed, Cochrane Central Register of Controlled Trials (CENTRAL), the WHO Global health library (filtering by the Index Medicus for the Eastern Mediterranean Region), Health Systems Evidence (HSE) and Google Scholar. We also searched Academic Search Complete, Directory of Open Access Journals, ScienceDirect, JSTOR journals, LexisNexis Academic: Law Reviews, World Bank eLibrary, and African Journals. We used free text terms and MeSH terms to search the different electronic databases for the following three concepts: digital health, fragile states and MENA region. S1 Appendix provides the search strategy of each database. We limited the search to articles published between 2010 and August 2022. We did not restrict the search to specific languages. We retrieved additional studies through screening the reference lists of relevant studies and contact of experts.

### Selection process

We imported the search results into Endnote X8 and removed duplicates. Two reviewers used the above eligibility criteria to screen titles and abstracts and full texts of identified citations for potential eligibility. They resolved any disagreements by discussion.

### Data charting

One reviewer abstracted data using standardized and pilot tested forms and another reviewer verified the abstracted data. Any disagreement between the initial abstractor and the verifier was resolved by discussion and when needed with the help of a third reviewer. We conducted calibration exercises to ensure the validity of the data abstraction process.

We abstracted the following information from each paper:

- Author information (last name of first author, country of the institution to which the first author is affiliated)

- Year of publication

- Type of study design (e.g., experimental, observational, literature reviews, systematic reviews, etc.)

- Country(ies) subject of the paper

  ○ Digital health interventions addressed in the study based on the targeted primary users (i.e. clients, healthcare providers, health system managers, data services)

- Type of technology employed

- Key findings that relate to the above interventions

## Data synthesis

We conducted descriptive analysis of the general characteristics of the included papers. We conducted a thematic analysis of the key findings of included studies categorized by targeted primary users of digital health intervention.

## Results

The PRISMA flowchart in Fig 1 summarizes the selection process. Out of the 10,724 articles identified through electronic database search, hand searching, and expert consultations, 93 met the inclusion criteria [20–113]. During the full-text screening, we excluded 100 articles

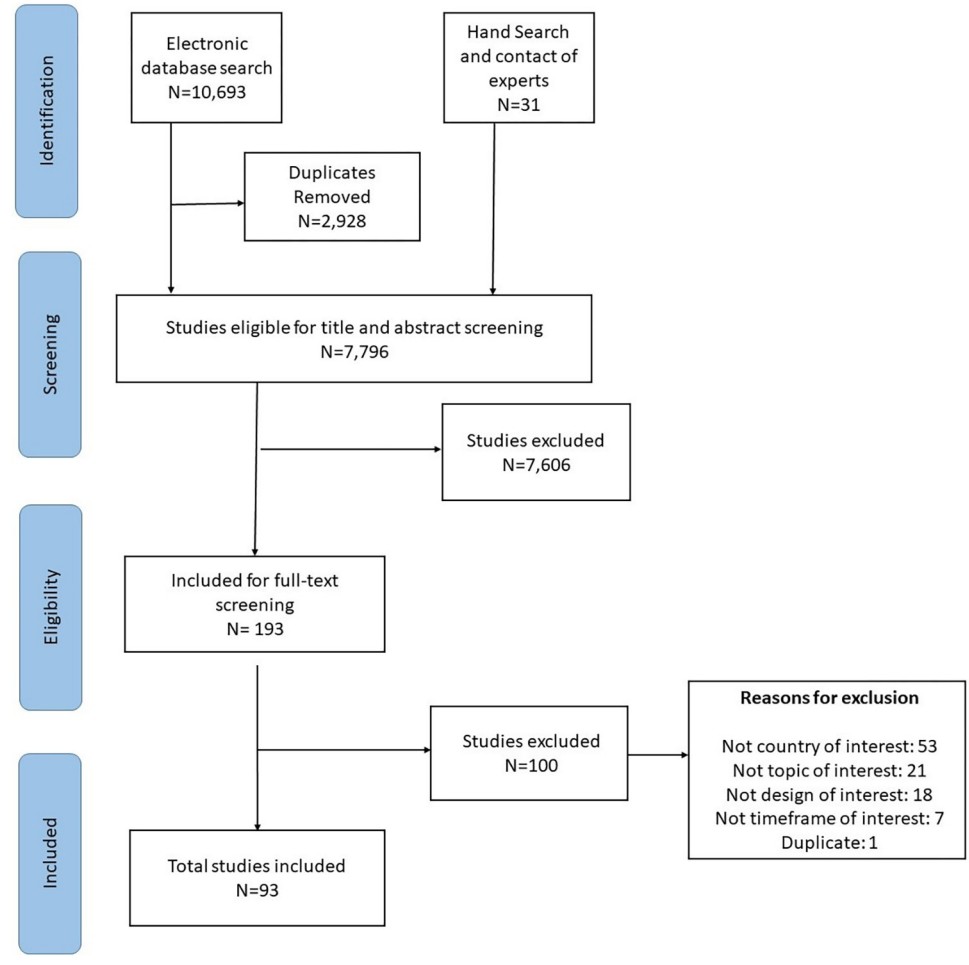

**Fig 1. PRISMA flowchart.**

based on the following reasons: not country of interest (n = 53), not topic of interest (n = 21), not design of interest (n = 18), not timeframe of interest (n = 7) and duplicate (n = 1) (S2 Appendix).

## Characteristics of included studies

Table 1 summarizes the characteristics of the 93 included studies. Most of the included studies were observational studies (n = 46; 49%), followed by descriptive case studies (n = 23; 25%), and experimental study designs (n = 12; 13%). Few studies employed quasi-experimental study design (n = 8; 9%) while three studies were a literature review and one was a modeling study. We identified no systematic reviews. Most of the studies were conducted in Lebanon (n = 30; 32%) followed by Afghanistan (n = 12; 13%), Palestine (n = 11; 12%), Sudan (n = 10; 11%), and Iraq (n = 10; 11%). The included studies focused on interventions targeting healthcare providers (n = 32; 34%), clients (n = 25; 27%), data services (n = 15; 16%), both healthcare providers and clients (n = 11; 12%), while fewer studies focused on health system or organizations managers (n = 10; 11%). The included papers mainly addressed topics on mhealth such as the use of mobile applications, smartphones, SMS and voice messages (n = 33; 35%), followed by telehealth/telemedicine that includes teleconsultations, teleradiology, tele-Pediatric (n = 32; 34%), and Electronic Medical Records (EMR)/Health Information Systems (HIS)/Surveillance Systems (n = 26; 28%). Digital health interventions were mainly employed to address non-communicable diseases including mental health (n = 28; 30%), sexual, reproductive, maternal, newborn, child, and adolescent health (n = 21; 23%) while fewer interventions addressed communicable diseases (n = 11; 12%). The first authors were mainly affiliated with institutions from countries outside the MENA region (n = 53; 57%), mainly the United Kingdom (UK) and United States (US). First authors affiliated with institutions from the MENA region were mainly affiliated with institutions from Lebanon. As shown in Fig 2, there is an increase in the production of evidence on digital health in fragile states of the MENA region over the period of 2010 to 2022 with a peak observed in 2021.

## Findings

We have organized the findings of these reviews based on the targeted primary users. Table of characteristics in supplementary files provide the characteristics on included studies organized by the targeted primary users.

**1. Healthcare Providers as Intended Users (n = 32):**

*a.* Thirty-two studies focused on digital health tools targeting healthcare providers namely physicians as primary intended users [26, 29, 34, 37, 38, 46–53, 57, 58, 61, 63, 64, 68–71, 74, 75, 80, 83, 89, 94, 96, 105, 108, 112] (S1 Table). Twenty-three studies (23/32) explored the effectiveness and implementation of digital health interventions aiming at supporting physicians through decision support algorithm [46, 47], telehealth decision support [37, 48–50, 68–70, 80, 83, 96, 108], mhealth decision support and medical record management [52, 53, 58, 71, 112] and e-learning [37, 57, 63, 64, 70, 75, 94] while nine studies (9/32) assessed health providers' readiness and willingness to adopt digital health tools [26, 29, 34, 38, 51, 61, 74, 89, 105].

*b. Decision support algorithms*

Two studies assessed the effectiveness of a digital algorithm tool for child illness management support in Afghanistan through quasi-experimental study designs [46, 47]. Both studies yielded positive results depicted in significant improvement in precautionary measures taken

**Table 1. Characteristics of included studies (N = 93).**

| Type of study design | | |
|---|---|---|
| | *N* | *%* |
| Observational studies (e.g. cross-sectional, qualitative, etc.) | 46 | 49% |
| Descriptive case studies | 23 | 25% |
| Experimental studies | 12 | 13% |
| Quasi-experimental studies | 8 | 9% |
| Literature reviews | 3 | 3% |
| Modeling studies | 1 | 1% |
| Systematic reviews | 0 | 0% |
| **Country subject of the paper** | | |
| | *N* | *%* |
| Lebanon | 30 | 32% |
| Afghanistan | 12 | 13% |
| Palestine | 11 | 12% |
| Sudan | 10 | 11% |
| Iraq | 10 | 11% |
| Somalia | 8 | 9% |
| Syria | 6 | 6% |
| Yemen | 2 | 2% |
| Djibouti | 2 | 2% |
| Libya | 2 | 2% |
| **Target users of digital health intervention** | | |
| | *N* | *%* |
| Healthcare providers | 32 | 34% |
| Clients | 25 | 27% |
| Data services | 15 | 16% |
| Clients & healthcare providers | 11 | 12% |
| Health system or organization managers | 10 | 11% |
| **Type of technology employed \*** | | |
| | *N* | *%* |
| Mhealth | 33 | 35% |
| *Clients* | 20 | 22% |
| *Providers* | 7 | 8% |
| *Clients & providers* | 3 | 3% |
| *Data services* | 2 | 2% |
| *Health managers* | 1 | 1% |
| Telehealth/telemedicine | 32 | 34% |
| EMR/HIS/Surveillance System | 26 | 28% |
| Not specified | 5 | 5% |
| Digital Algorithms | 2 | 2% |
| **Disease conditions** | | |
| | *N* | *%* |
| Not Specified disease condition | 33 | 35% |
| Non-Communicable Diseases *(including mental health; n = 17)* | 28 | 30% |
| Sexual, reproductive, maternal, newborn, child, and adolescent health | 21 | 23% |
| Communicable Diseases (COVID-19; n = 4) | 11 | 12% |
| **Country of the institution to which the first author is affiliated** | | |
| | *N* | *%* |

(*Continued*)

**Table 1.** (Continued)

| Countries outside the MENA region | 53 | 57 |
|---|---|---|
| United States | 10 | 11% |
| UK | 10 | 11% |
| Norway | 6 | 6% |
| Switzerland | 4 | 4% |
| France | 4 | 4% |
| Kenya | 3 | 3% |
| The Netherlands | 3 | 3% |
| Germany | 2 | 2% |
| Malaysia | 2 | 2% |
| Other countries | 9 | 10% |
| Fragile states from the MENA region | 34 | 37% |
| Lebanon | 18 | 19% |
| Iraq | 5 | 5% |
| Sudan | 4 | 4% |
| Palestine | 3 | 3% |
| Libya | 2 | 2% |
| Somaliland | 1 | 1% |
| Yemen | 1 | 1% |
| Other countries from the MENA region | 6 | 6% |
| Kingdom of Saudi Arabia | 3 | 3% |
| Pakistan | 3 | 3% |
| **Reporting of Funding** | | |
| | *N* | *%* |
| Yes | 64 | 69% |
| No | 29 | 31% |
| **Reporting of Conflict of Interest** | | |
| | *N* | *%* |
| Yes | 71 | 76% |
| No | 22 | 24% |

[46], appropriateness of physical examination [47] and treatment [47] along with a significant decrease in antibiotic prescription [46, 47].

*c. Telehealth for providers decision support*

Seven observational studies [37, 48–50, 68, 70, 80] and four case studies [69, 83, 96, 108] described the implementation of telehealth for physician decision support. Telehealth was employed to address the shortage of physicians resulting from their migration from conflict-affected countries such as Syria [69, 83], and the shortage of specialists in low resource settings such as Djibouti [48, 49]. Other purposes of telehealth included supporting local staff in war-torn countries such as Somalia [68, 80, 108], supporting a family medicine education program in Sudan [37, 70], and assessing the feasibility of tele-ophthalmology within the Lebanese setting [50].

The majority of these studies devised basic technologies such as websites, emails, social media platforms, digital cameras, and webcams for information exchange and decision support [37, 48–50, 68–70, 80, 83] supported by electronic medical records to facilitate exchange [37, 70].

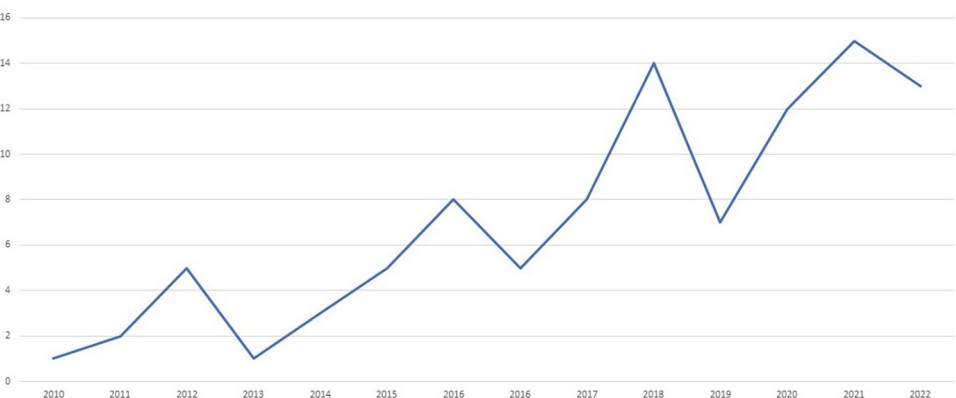

**Fig 2. Annual production of evidence on digital health in fragile states in the MENA region.**

These interventions supported curative care such as pediatric care [80], pediatric intensive care [83], pediatric orthopedics [48], orthopedic surgery [49], hospital care [68], family medicine [37, 70] and diagnostic care such as radiology [69] and ophthalmology [50].

These interventions were perceived as useful [80] and important [37], and resulted in positive outcomes; such as improving case management [48, 80], detection of diseases [50, 80], and survival rates [83] resulting in very good clinical outcomes [48, 49], and acceptable levels of adverse outcomes [68] resolving diagnostic uncertainty [48, 49], and improving physicians' skills [80]. Yet, one study reported challenges in teleradiology due to poor CT-Scan and uploaded pictures quality [69].

*d. mhealth for decision support and medical record management*

Two observational studies conducted in Lebanon assessed the effectiveness of an intervention including the implementation of disease management guidelines in addition to a mhealth tool for decision support and medical record management in improving the quality of noncommunicable disease management for Syrian refugees and vulnerable Lebanese in primary care setting [52, 53]. The first study (pilot study) noted an overall low reporting and low uptake yet a significant improvement in BMI and blood pressure reporting in the mHealth application as compared to clinic medical records (47.4% vs 15.8%, p<0.001 and 74.5% vs. 40.7%, p<0.001 respectively); and more frequent reporting of weight, height, blood pressure, and blood sugar measurement; and improved patient-provider interaction [52]. Similarly, for the second study the rate of blood sugar, blood pressure, weight, height and BMI recording improved but not significantly except for BMI (8.1%, P < .001). The intervention also resulted in an improvement in the frequency of lifestyle counseling and the quality of patient-provider interaction [53]. The two studies reported the poor connectivity as a main challenge to the use of the application.

A study conducted in Palestine described the implementation of a mhealth tool (open data kit) for child mental illness screening. A total of 986 children were screened and the authors attributed part of the intervention's success to the tool's capability to collect data offline especially in difficult conditions in war-torn areas [58]. Another study conducted in Lebanon, demonstrated the effectiveness of an e-health tool (netbook) in screening non-communicable diseases namely hypertension and diabetes, and automating appointment scheduling, and referrals in outreach activities for underserved communities [71].

*e. E-learning*

Five observational studies [37, 63, 64, 70, 94], a case study [57], and a quasi-experimental study assessed and/or described the implementation of e-learning [37, 57, 63, 64, 70] and blended learning programs [75]. The programs included peer-to-peer learning between students in the UK and students in Somaliland [57, 63, 64] for real-time case-based bedside training [57], and psychiatry training [63, 64], use of social media for COVID-19 training [94], family medicine education in Sudan [37, 70], and health providers capacity building in rural Afghanistan [75].

The technologies devised for e-learning were mainly websites for instant messaging [63, 64], information exchange [57], and virtual tutoring in virtual classrooms [37, 57, 70], supported by electronic medical records [37, 70].

Overall, the providers were satisfied with e-learning interventions [37, 57, 63] and showed improvement in knowledge [64, 75], yet one study reported challenges related to scheduling and coordination [64].

*f. Assessment of healthcare providers' readiness/willingness to adopt digital health*

Six observational studies [26, 29, 38, 51, 61, 74] and one modeling study [34] assessed health workers readiness/willingness to adopt digital health tools such as telemedicine [29, 38, 51, 61] and health information systems [26, 34] and the influencing factors that affect their acceptance.

Three studies were conducted in Lebanon [29, 51, 74, 105], two were conducted in Iraq [34, 38], one in Palestine [26], and one in Syria [61]. Two studies assessed physicians' perception and use of telehealth in Lebanon. Physicians reported frequent use of various phone applications, but the majority only perceived social media and telehealth as useful tools for communications among physicians. Less than half of the respondents believed that mhealth and telehealth can be used to manage patient health by supporting patient education and treatment adherence. Physicians also voiced concerns related to patient privacy and medicolegal issues [51]. During the COVID-19 pandemic, a significant improvement in telehealth use was observed for various functions such as discussing medical cases with patients or prescribing medications. Additionally, physicians were more likely to perceive the importance of digital health in the post-pandemic era and reported its ease of use [29]. A study conducted in Lebanon assessed providers' ehealth readiness in primary healthcare facilities and indicated the readiness of healthcare providers to adopt digital health [74]. The two studies conducted in the Iraqi setting identified influencing factors affecting physicians' willingness to utilize telehealth [38] and health information systems [34]. Common factors across both studies included privacy and system compatibility [34, 38]. The study conducted in the Palestinian setting assessed nurses' readiness to adopt electronic health records. Overall, their attitude was positive and showed a positive correlation with highest degree obtained [26]. The study conducted in the Syrian setting assessed providers' perception of tele-mental health. Although the majority of providers didn't have experience with the technology they still believed that it could be of benefit to their patients [61].

## 2. Clients as Intended Users (n = 25):

Twenty-five studies addressed clients as primary intended users [21, 22, 25, 27, 30, 32, 33, 42, 45, 60, 67, 72, 73, 77, 78, 82, 84, 86, 87, 90, 93, 95, 97, 99, 104] (S2 Table). These studies assessed the effectiveness digital health interventions in eliciting behavioral change related to patient adherence to treatment and described their implementation [21, 27, 30, 32, 33, 42, 60, 67, 72, 73, 77, 86, 87, 93, 95, 97]. and assessed population's readiness to adopt digital health tools [36, 45], factors affecting the adoption [22, 78], and their perception of mhealth interventions [25].

*a. Digital health interventions for behavioral change and adherence to treatment*

Nine experimental studies [21, 30, 32, 42, 60, 67, 72, 95] and one observational study [73] assessed the effectiveness of mhealth tools including mobile applications [20], text [21, 30, 32, 42, 67, 72, 73] and voice messages [30], and a mhealth system including an application connected via Bluetooth to a glucose monitor [60]. These interventions aimed at improving hypertension [20, 72, 95] and diabetes management [60], eliciting lifestyle changes for NCD management [73], enhancing adherence to tuberculosis treatment [21], antenatal care visits [42], mammography screening [67], and contraceptive use [32], and improving childcare [30]. The majority of the studies yielded positive outcomes demonstrated in an improvement in health outcomes such as HBA1c levels [60, 72], and blood pressure control [72], behavior modification [73], improvement in tuberculosis cure rate [21], and improvement in adherence to treatment for hypertension [20], adherence to antenatal care visits [42], increased odds of contraception use [32], and improvement in knowledge around maternal and child care [30]. Additionally, results of the experimental study conducted by Lakkis et al. indicated that both brief and longer SMS were equally effective (30.7% and 31.6% respectively) in encouraging women to do their routine mammography [67]. One study on using text messages suggested coupling voice messages with text messages to assist for illiterate individuals [73].

Six studies assessed the effectiveness of digital health interventions (e.g. mobile health, websites) [33, 77] for the provision of mental health services for conditions such as depression [33, 77, 86, 87, 93, 97], anxiety [77], and post-traumatic stress disorder [77]. Both studies demonstrated an improvement in depression symptoms, improvement in the Wilcoxon signed ranks test [33], drop in total scores for Posttraumatic Diagnostic Scales (PDS), reduction in symptoms on the HSCL-25 depression and anxiety subscales, increase in quality of life, and decrease in percentage of patients with clinically significant complaints [77]. In the study conducted in Lebanon, patients were satisfied with their relationship with the non-specialist support person (e-helper), yet e-helpers noted the need for more training on complex cases [33].

Finally, one observational study described the implementation of a targeted client communication (TCC) intervention using text messages for maternal and child health [27]. The intervention was co-designed with users based on behavior change theories, knowledge and awareness gaps were identified, and messages were tailored based on these knowledge gaps in addition to the patient's identified risk in electronic health records [27].

*b. Assessment of clients' readiness to adopt digital health tools, factors affecting their adoption, and their perception*

Two observational studies assessed users' readiness to adopt mhealth tools for maternal and child care in Afghanistan and mental health in Palestine [45, 78]. Results of the study showed that the majority of the respondents owned a mobile phone, were familiar with their services, were open to receiving SMSs and reminders and to call a helpline, and agreed that mhealth can be used to support health [78]. The study found that literacy rate might be a factor influencing mhealth adoption [78]. The second assessed the viability of using mhealth approaches as an alternative to traditional mental health services in rural areas in Palestine had internet access, and were willing to use mHealth tools for mental health support [45].

A literature review [22] and an observational study [82] were conducted to identify factors influencing mhealth adoption by older adults in Iraq [22] and digital health adoption by refugees in Lebanon [82] respectively. The literature review identified perceived usefulness, ease of use, and subjective norm as predictors of mobile health adoption by older adults and perceived usefulness, subjective norm, and facilitating conditions as predictors of their intention to use mobile health [22]. The study designed to identify predictors of refugee digital health use

identified key considerations that should be taken into account while designing the technology including: (1) refugees' health believes and experiences; (2) their literacy level, (3) their perceptions during previous encounters with providers, and (4) their hierarchal, cultural and familial structures [82].

Finally, one observational study assessed lay people's perception of six mobile health applications for weight management [25]. Users assessed these applications as highly functional yet not engaging, and the applications' subjective quality scores were low indicating that the users' intentions to reuse was low [25].

### 3. Data services (n = 15)

Fifteen studies examined the use of digital health in supporting activities related to data services such as data collection, management and analysis [23, 41, 43, 44, 55, 56, 59, 62, 76, 85, 100, 102, 103, 111, 113] (S3 Table). Ten studies discussed how the use of digital health can improve health information systems [43, 44, 55, 76, 85] and disease surveillance [23, 62, 100, 111, 113] in conflict and low resources settings. The transition from paper-based health information to eRegistry for maternal and child health in West Bank showed promising results in generating more reliable and complete health systems indicators [76]. The limited use of digital health is a main weakness of health information systems in Somaliland [43] and Lebanon [44]. One study recommended the use of mobile community-based information system for data collection to strengthen the health information system in Somaliland [43] while the other recommended the use of the internet and social media to facilitate data retrieval and information sharing [44]. A study in Sudan assessing the readiness of the health information system for the digitization found challenges related to information and communication technology infrastructure, capacity building and coordination that need to be addressed [55].

Five studies assessed how the use of digital health tools such as smartphones can enhance data collection in health research and surveys and for planning immunization campaigns [41, 59, 102, 103] and can facilitate data collection in conflict settings to inform the work of humanitarian agencies [56].

### 4. Healthcare Providers and Clients as Intended Users (n = 11)

Eleven studies assessed the feasibility, effectiveness, and user perception of interventions targeting both healthcare providers and clients as intended users [24, 28, 35, 39, 40, 65, 66, 88, 98, 101, 107] (S4 Table). Five of these studies assessed interventions aiming at leveraging technology to provide virtual care [35, 39, 40, 88, 98] while two studies assessed a multi-component intervention aiming at mobilizing different technologies for e-learning and mental health screening and awareness [65, 66]. Additionally, two studies assessed patients and providers' perception of electronic medical records [24] and electronic patients portals [28].

#### a. Virtual Care

Two studies assessed telehealth interventions including tele-psychiatry [39] in Somalia, and telemedicine for home palliative care [35] in Lebanon. In Somalia, patients at a local clinic connected with Somali psychiatrists abroad through Skype [39]. Through tele-psychiatry, patients were examined and diagnosed with mental illnesses such as schizophrenia and psychosis [39]. In Lebanon, a palliative care center utilized telehealth including phone calls, video-calls, and messaging to support caregivers of patients outside its geographical reach [35]. Both caregivers and providers positively perceived the intervention yet some caregivers still preferred face-to-face support [35].

Another observational study contextualized an e-mental health intervention to the Lebanese setting [40] by shortening its content, adding a video representation, increasing the focus

on mood lifting activities, and adapting the mode of contact. Two studies examined the use of telehealth to provide services during the COVID-19 pandemic [88, 98].

*b. E-learning, Awareness, and Screening*

Two studies described and assessed the effectiveness of a multi-component community-based intervention [65, 66]. The objective of the intervention was to improve awareness on mental illness among young adults in Afghanistan through educational SMS and to support community workers in diagnosis and reporting through a mobile application that enabled them to record patient information and access learning material and guidelines [65, 66]. The results of the quasi-experimental study demonstrated that the intervention was successful in improving awareness about mental health in the community and among young adults, reducing stigma in the community and improving referrals to mental health services [66].

*c. Health Information Systems*

Two observational studies were conducted to assess clients and providers perception of health information systems in Lebanon [24, 28]. In the first study, patients reported that the use of electronic medical records did not negatively affect their communications with their providers [24]. The second study assessed clients and providers' acceptance of electronic medical portals. Providers reported improved perceived ease of use and more privacy concerns than patients, yet both providers and patients reported high perceived usefulness [28].

5. **Health system or organizations managers as intended users (n = 10)**

Ten studies addressed health system, organizations or resource managers [31, 36, 54, 79, 81, 91, 92, 106, 109, 110] (S5 Table). Three study assessed the role of digital health in and improving service access, quality and efficiency [81, 109, 110], four assessed e-health readiness [36, 54, 91, 92, 106], three described the implementation of health monitoring and surveillance systems [31, 79, 106].

*a. Digital health for access, quality and efficiency*

One study conducted in Yemen assessed rural areas' challenges to improve health care access and quality and identified challenges related to services (lack of facilities and bad transportation), human resources (lack of continuing education and high rate of medical errors), and health management information (lack of health information system). The findings showed that telehealth might be an effective tool in meeting those needs (for instance for e-learning) [81]. Two studies in Palestine reported on the effectiveness of digital health registry in improving the quality and efficiency of health services [109].

*b. ehealth readiness*

Four studies assessed eHealth readiness in two healthcare institutions in Afghanistan [54], and hospitals [91] primary care centers in Lebanon [36]. The study conducted in Afghanistan identified the provision of care, capacity building and information management as prerequisites for successful digital health interventions' implementations [54]. The second study assessed the primary care centers' readiness to integrate health technology for the provision of maternal and child health services for Syrian refugees in Lebanon. It identified key challenges including the unavailability of time and technological resources in some centers, the interviewees' perception of refugee technological illiteracy, and their hesitation to invest in technology for a population they perceived as mobile [36]. The cost, lack of legislation and personnel resistance and the workload were also reported challenges to implementation of ehealth in hospitals in Lebanon [91, 92].

*c. Health Monitoring and Disease Surveillance*

Three case studies described the implementation of a health monitoring system in Iraq [31, 106] and a disease surveillance system in Sudan [79]. The Iraqi study described the successful implementation of a health monitoring system in 59 primary care centers. The authors highlighted the importance of establishing multiple stakeholder partnerships for the success of such project implementations [31]. The study conducted in Sudan described the implementation of a cheap SMS application for weekly surveillance reporting. The users perceived this tool as interesting and easy to use, and the timeliness of data reporting improved. Yet, users identified implementation challenges such as poor network coverage in some centers and the unavailability of dedicated staff in others [79].

## Discussion

Digital health technologies are used for a wide variety of health systems functions and for the benefit of different actors. Many of these technologies show promising solutions in addressing health needs in fragile and conflict-affected states. As per the findings of this scoping review, they provide a platform for training, supervision, and consultation for health care providers, continuing education for medical students, and disease self-management in addition to increasing access to health care and health information and improving the ability to diagnose, treat, and track diseases. However, the adoption of digital health is not without challenges. Findings of some of the included papers also showed a raised concern about privacy among patients and providers. Additional challenges include limited computer literacy, unavailability of resources (e.g. staff), weak technological infrastructure and network coverage, restrictive cultural norms and practices, and time burdens. Specific challenges on the use of mhealth included poor connectivity, privacy and illiteracy concerns and illiteracy. These findings concur with another systematic review on the application of m-health technologies in developing countries that stated several limitations, which included interoperability and lack of a technology infrastructure [114].

To successfully adopt and implement digital technologies in fragile settings a number of factors should be taken into consideration. First, the development and design of digital technologies in these settings need to gain a deeper understanding of the target population's health beliefs, culture, context, and use of technology in addition to the issue of privacy. To ensure privacy, security, and other ethical issues in the use of digital technology, a regulatory body, whether independent or governmental, should be established to supervise these technologies. The availability of the resources and willingness of communities and healthcare providers are also key in assessing the health systems readiness and response for adoption of digital technologies [115]. Assessing readiness of health care providers and clients is key to ensure successful implementation of any digital health [74]. The acceptance and reception of health providers of digital technologies can be improved through the introduction of educational courses on digital health in medical educational curriculum and through mandating core competencies in this field for public health decision makers and managers of health organizations. Additionally, it is very important to assess the health needs and priorities in a specific setting a priori to the introduction of a digital health technology to ensure high return on investment and that the intervention is being delivered to the area and people most in need.

## Strengths and limitations

To our knowledge, this is the first study to map out the evidence on digital health in fragile states in the MENA region. One strength is that we followed JBI guidance for conducting

scoping reviews [116] and the PRISMA Extension for Scoping Reviews (PRISMA-ScR) for reporting scoping reviews [16]. One limitation of this review is that we did not search websites of humanitarian organizations working in fragile and conflict-affected settings.

### Research gaps and implication for future research

This scoping review showed the increase of production of research studies on digital health use in fragile states in the MENA region. The review found limited experimental and quasi-experimental studies highlighting the limited evaluation and impact assessment research on this topic. The review also highlighted the lack of systematic reviews synthesizing evidence on the effectiveness of digital health interventions in fragile states of the MENA region. Future research should focus on evaluation and impact assessment studies and systematic reviews of effectiveness in order to gain a better understanding on the effectiveness and cost-effectiveness of different digital health technologies. Rigorous evaluation of digital technologies in humanitarian crisis are warranted to inform their design and deployment across different settings. These findings are in line with a review [6] that highlighted the need for evaluative efforts that support data integrity, protection, and security across all stages of the data life cycle. Researchers are also called to address gender equity in their methodology and analysis as highlighted in a recent review that found that few digital health projects implemented in Africa, Asia, Latin America and the Middle East encompass gender analysis and health equity in their methodologies [117]. And, in light of the multiple vulnerabilities and experiences that can disadvantage individuals and groups in conflict settings, adopting an intersectional analysis for the needs assessment and subsequent processes would be required [118].

The findings of this review also highlighted the limited of digital health interventions targeting health system or organization managers such as interventions targeting managerial functions related to supply chain management, health financing, human resource management. This finding showed the need to explore the potential of digital health in improving health system functioning. As most of the first authors were affiliated with institutions based in countries outside the MENA region, this scoping review might reflect the limited capacity and resources in this region to conduct health research.

### Supporting information

**S1 Checklist. Preferred Reporting Items for Systematic reviews and Meta-Analyses extension for Scoping Reviews (PRISMA-ScR) checklist.**
(PDF)

**S1 Appendix. Search strategies.**
(DOCX)

**S2 Appendix. List of excluded articles with reason for exclusion.**
(DOCX)

**S1 File. Data abstraction from.**
(XLSX)

**S1 Table. Studies on health providers as intended users.**
(DOCX)

**S2 Table. Studies on clients as intended users.**
(DOCX)

**S3 Table. Studies on data services.**
(DOCX)

**S4 Table. Studies on health providers and clients.**
(DOCX)

**S5 Table. Studies on health systems or organizations managers.**
(DOCX)

## Author Contributions

**Conceptualization:** Fadi El-Jardali, Lama Bou-Karroum, Patricia Mecheal, Chaitali Sinha.

**Data curation:** Lama Bou-Karroum, Mathilda Jabbour, Karen Bou-Karroum, Andrew Aoun, Sabine Salameh.

**Formal analysis:** Lama Bou-Karroum, Mathilda Jabbour, Karen Bou-Karroum, Andrew Aoun.

**Funding acquisition:** Fadi El-Jardali.

**Methodology:** Lama Bou-Karroum.

**Project administration:** Fadi El-Jardali, Lama Bou-Karroum.

**Supervision:** Lama Bou-Karroum.

**Validation:** Lama Bou-Karroum, Patricia Mecheal, Chaitali Sinha.

**Writing – original draft:** Lama Bou-Karroum, Mathilda Jabbour, Karen Bou-Karroum, Andrew Aoun.

**Writing – review & editing:** Fadi El-Jardali, Lama Bou-Karroum, Mathilda Jabbour, Karen Bou-Karroum, Andrew Aoun, Sabine Salameh, Patricia Mecheal, Chaitali Sinha.

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
