## [Decision Letter · Decision Letter 0]

22 Feb 2023

PONE-D-22-35597Digital Health in Fragile States in the Middle East and North Africa (MENA) Region: A Scoping Review of the LiteraturePLOS ONE

Dear Dr. El-Jardali,

Thank you for submitting your manuscript to PLOS ONE. After careful consideration, we feel that it has merit but does not fully meet PLOS ONE’s publication criteria as it currently stands. Therefore, we invite you to submit a revised version of the manuscript that addresses the points raised during the review process.

We look forward to receiving your revised manuscript.

Kind regards,

Sebastien Kenmoe

Academic Editor

PLOS ONE

Journal Requirements:

  "This work is supported by the International Development Research Centre in Canada. The funding has ended." 

   "none to declare"

5. Please ensure that you include a title page within your main document. You should list all authors and all affiliations as per our author instructions and clearly indicate the corresponding author.

6. Please include a caption for figure 2.

Additional Editor Comments:

It is also important for Authors to justify why they focused their search from 2010.

Reviewers' comments:

Reviewer's Responses to Questions

**Comments to the Author**

1. Is the manuscript technically sound, and do the data support the conclusions?

Reviewer #1: Yes

2. Has the statistical analysis been performed appropriately and rigorously? 

Reviewer #1: N/A

3. Have the authors made all data underlying the findings in their manuscript fully available?

Reviewer #1: Yes

4. Is the manuscript presented in an intelligible fashion and written in standard English?

Reviewer #1: Yes

5. Review Comments to the Author

Reviewer #1: This scoping review covers all digital health interventions in MENA from 2010 - 2022.

This manuscript is well-written and adds important findings to the literature.

I have only few suggestions.

The introduction is shorter than I would have anticipated.

Page 9: The included papers mainly addressed topics on mhealth (n=34; 36%), followed by telehealth/telemedicine (n=32; 34%), and EMR/HIS/Surveillance Systems (n=26; 28%).

What distinction are you making btwn these categories above ? It’s not clear

I very much like the analysis of home country of author but I would put information about “disease conditions” before this information both in the table and in the narrative. Share the information about the intervention then share about author country institution.

Would also like more detail about topic of the intervention – the categories as-is are fairly broad.

Page 12, clarify that you’re talking about 19 of the 32.

Page 12, list the results in the opening in the order you’re going to present them in subsection a – e

Throughout results, be more specific. For example “The first study (pilot study) noted an overall low reporting and low uptake yet an improvement in BMI and blood pressure reporting; weight, height, blood pressure, and blood sugar measurement; and patient-provider interaction [49].” -what does low reporting and low uptake mean in this context? Sharing more quantitative results will strengthen the paper. And again on page 15 “Overall, the providers were satisfied with e-learning interventions [34, 54, 60] and showed improvement in knowledge [61, 72], yet one study reported challenges related to scheduling and coordination [61].” – could you quantify any of the improvements in knowledge?

It would be nice to have a table that breaks down the topic of the mhealth intervention by target audience

“The objective of the intervention was to improve awareness on mental illness among young adults in Afghanistan through educational SMS” – how many of the articles you reviewed did NOT require a smartphone (when the audience was clients)?

Did any of the articles talk about cell phone or internet limiations, and for articles that were client-facing, the limitations of cell phone ownership in the population? This might be worth addressing in the discussion section. Especially for those who are less familiar with the MENA region, it might be helpful to give the context of whether cell phones are ubiquitous or not, and if not, who doesn’t have them (thus who could not be reached for certain client-facing interventions).

What other implementation issues were there?

Discussion section: page 23 “The implementation of digital health raises concerns such as curtailing privacy of individuals and groups.” – was this addressed throughout the narrative? If not, this isn’t a finding of the scoping review and rather should be placed in a different section of the discussion section – an area that focuses on further considerations, perhaps.

Page 25 – 2nd line – type-o (repeated word)

6. PLOS authors have the option to publish the peer review history of their article (what does this mean?). If published, this will include your full peer review and any attached files.

Reviewer #1: No

---

## [Author Response · Author response to Decision Letter 0]

4 Apr 2023

Editor comments

Comment 1: Please ensure that your manuscript meets PLOS ONE's style requirements, including those for file naming. The PLOS ONE style templates can be found at https://journals.plos.org/plosone/s/file?id=wjVg/PLOSOne_formatting_sample_main_body.pdf and https://journals.plos.org/plosone/s/file?id=ba62/PLOSOne_formatting_sample_title_authors_affiliations.pdf

Response 1: we revised te PLOS ONE's style requirements, including those for file naming (for figures and tables) and we made sure to comply with the journal requirements. 

Comment 2: Thank you for stating the following financial disclosure: 

 "This work is supported by the International Development Research Centre in Canada. The funding has ended." 

Response 2: We confirm that the funder, “the International Development Research Centre” (IDRC) in Canada, had no role in the study and we included a statement on the role of funder in the manuscript: "The funders had no role in study design, data collection and analysis, decision to publish, or preparation of the manuscript." Chaitali Sinha contributed to the conceptualization and reviewed the final draft of the manuscript. 

Comment 3: Thank you for stating the following in your Competing Interests section: 

 "none to declare"

Response 3: We confirm that all authors have no competing interests as mentioned in the manuscript. No changes were made to this statement. 

Comment 4: In your Data Availability statement, you have not specified where the minimal data set underlying the results described in your manuscript can be found. PLOS defines a study's minimal data set as the underlying data used to reach the conclusions drawn in the manuscript and any additional data required to replicate the reported study findings in their entirety. All PLOS journals require that the minimal data set be made fully available. For more information about our data policy, please see http://journals.plos.org/plosone/s/data-availability.

Response 4: the tables in the supplementary files include results of individual studies underlying the results. As per PLOS One requirements, we also uploaded the data abstraction form that includes all the data set underlying the results described in our manuscript as Supporting Information file during the resubmission. 

Comment 5: Please ensure that you include a title page within your main document. You should list all authors and all affiliations as per our author instructions and clearly indicate the corresponding author.

Response 5: the main document includes a tile page that list all authors and all affiliations. We have updated the title page to indicate the corresponding author with email address (Fadi El-Jardali, fe08@aub.edu.lb). 

Comment 6: Please include a caption for figure 2.

Response 6: We included the below caption for figure 2 in the manuscript: “Annual production of evidence on digital health in fragile states in the MENA Region”.

Comment 7: Please include captions for your Supporting Information files at the end of your manuscript, and update any in-text citations to match accordingly. Please see our Supporting Information guidelines for more information: http://journals.plos.org/plosone/s/supporting-information. 

Response 7: we included captions for the supporting information files at the end of the manuscript as requested:

Supporting files

S1 Appendix. Search strategies

S2 Appendix. List of excluded articles with reason for exclusion 

S3 Table. Studies on health providers as intended users

S4 Table. Studies on clients as intended users

S5 Table. Studies on data services

S6 Table. Studies on health providers and clients

S7 Table. Studies on health systems or organizations managers

Comment 8: Please review your reference list to ensure that it is complete and correct. If you have cited papers that have been retracted, please include the rationale for doing so in the manuscript text, or remove these references and replace them with relevant current references. Any changes to the reference list should be mentioned in the rebuttal letter that accompanies your revised manuscript. If you need to cite a retracted article, indicate the article’s retracted status in the References list and also include a citation and full reference for the retraction notice.

Response 8: We revised all references and we made the necessary changes to make sure they are correct and complete. We found on article that was duplicate (reference no. 17 and reference no. 92 in the old version of the manuscript). We captured the same article during the update where it was cited to be published in 2021 while we had it cited as 2020 in our initial list of references. Therefore, we changed the total number of included studies to 93 and we added one study to the excluded list:

The PRISMA flowchart in Figure 1 summarizes the selection process. Out of the 10,724 articles identified through electronic database search, hand searching, and expert consultations, 93 met the inclusion criteria [20-113]. During the full-text screening, we excluded 100 articles based on the following reasons: not country of interest (n=53), not topic of interest (n=21), not design of interest (n=18), not timeframe of interest (n=7) and duplicate (n=1) (S2 Appendix). 

We also revised all the in-text citation to made sure all the references cited in the paragraph opening in the results section are also cited in the sub-sections. We also revised all the reference list and we do not have any retracted articles cited.

Additional Editor Comments:

It is also important for Authors to justify why they focused their search from 2010.

Response: As our review focus on the MENA region, we restricted the eligibility to 2010 as this region experienced significant changes after this year. The main event that took place in the region is the Arab Spring which began in Tunisia in 2010 and sparked the protests and uprisings throughout the region. The protests in many countries turned into devastating conflict mainly in Syria (2011), Libya (2011) and Yemen (2014). In addition, the field of digital health started to witness major changes with the widespread use of smartphones that increased significantly after 2010. We have added the below to the methods section under the eligibility criteria sub-section:

Setting of interest: We included studies on fragile states from the MENA region. We considered fragility caused by conflicts as well as institutional fragility. As per the World Bank 2019 classification of fragile and conflict-affected situations, the fragile states in the MENA region are: Afghanistan, Djibouti, Iraq, Lebanon, Libya, Somalia, Sudan, Syria, West bank and Gaza, and Yemen [19]. We restricted our eligibility to articles published after the year 2010 as this region experienced significant changes after this year mainly the Arab Spring which began in 2010 and sparked the protests, uprisings and conflict throughout the region.

Reviewer Comments to the Author

Reviewer #1

Comment 1: This scoping review covers all digital health interventions in MENA from 2010 - 2022. This manuscript is well-written and adds important findings to the literature.

I have only few suggestions.

Response 1: We thank the Reviewer for the positive feedback. We worked to address all your suggestions to improve the quality of our manuscript. 

Comment 2: The introduction is shorter than I would have anticipated.

Response 2: We have revised the introduction section to make sure it includes more about existing evidence on digital health and to strengthen the rational of our scoping review:

The COVID-19 pandemic highlighted the importance of digital health in effectively responding and containing this public health crisis [10, 11]. In low and middle-income countries, digital health, in particular telemedicine, was found to reduce transmission risk and improve access to healthcare during COVID-19 [12]. Despite the reported benefits of locally relevant and appropriately designed digital technologies, they are still underutilized in fragile states in the MENA region. Digital health research is also still in its infancy in this region with evident disparities in comparison to the global context [13]. To inform future research, investments and policy, the objective of this scoping review is to map out the evidence on digital health in fragile states in the MENA region. To the best of our knowledge this is the first scoping review to focus on the region. Previous reviews focused on the role of digital health in humanitarian settings [14] or in conflict-affected population [6] in general with no specific focus on the MENA region. 

Comment 3: Page 9: The included papers mainly addressed topics on mhealth (n=34; 36%), followed by telehealth/telemedicine (n=32; 34%), and EMR/HIS/Surveillance Systems (n=26; 28%). What distinction are you making btwn these categories above? It’s not clear

Response 3: We categorized the included studies by type of digital health as reported in the paper itself and then we regrouped them by mhealth, telehealth/telemedicine, and EMR/HIS/Surveillance Systems. For instance, if the study report on mobile health, use of smartphones, SMS, mobile applications, we categorize it under mhealth. Studies that report on teleradiology, tele-Pediatric, teleconsultations were categorized under telemedicine. Studies reporting on electronic medical records, health information systems and surveillance systems were grouped together as in many instances we found that they were inter-related. This categorization is in line with a book on digital health pubished in 2022 that clearly separate between these 3 categories .

Comment 4: I very much like the analysis of home country of author but I would put information about “disease conditions” before this information both in the table and in the narrative. Share the information about the intervention then share about author country institution.

Response 4: We thank the Reviewer for the positive feedback on the analysis of home country of author. We agree with the above comment and we have put the information on disease conditions before the home country of author in the table of characteristics and in the narrative section of “Characteristics of included Studies”. 

Comment 5: Would also like more detail about topic of the intervention – the categories as-is are fairly broad.

Response 5: As mentioned in the response 3, we categorized the included studies by digital health intervention as reported in the paper itself and then we regrouped them by mhealth, telehealth/telemedicine, and EMR/HIS/Surveillance Systems. To better clarify the categories for the readers, we added the below to the results section:

The included papers mainly addressed topics on mhealth such as the use of mobile applications, smartphones, SMS and voice messages (n=34; 36%), followed by telehealth/telemedicine that includes teleconsultations, teleradiology, tele-Pediatric (n=32; 34%), and Electronic Medical Records (EMR)/Health Information Systems (HIS)/Surveillance Systems (n=26; 28%).

Comment 6: Page 12, clarify that you’re talking about 19 of the 32.

Response 6: as we have revised the opening paragraph to better represent the sub-sections, we also made sure to clarify the numbers by adding the denominators:

Thirty-two studies focused on digital health tools targeting healthcare providers namely physicians as primary intended users [26, 29, 34, 37, 38, 46-53, 57, 58, 61, 63, 64, 68-71, 74, 75, 80, 83, 89, 94, 96, 105, 108, 112] (S3Table). Twenty studies (23/32) explored the effectiveness and implementation of digital health interventions [46-50, 52, 53, 61, 64, 75, 80, 83][57, 58, 63, 68-71] aiming at supporting physicians through decision support algorithm [46, 47], telehealth decision support [37, 48-50, 68-70, 80, 83, 96, 108], mhealth decision support and medical record management [52, 53, 58, 71, 112] or and e-learning [37, 57, 63, 64, 70, 75, 94] while nine studies (9/32) assessed health providers’ readiness and willingness to adopt digital health tools [26, 29, 34, 38, 51, 61, 74, 89, 105][29, 38, 51, 61][26, 34][74]. 

Comment 7: Page 12, list the results in the opening in the order you’re going to present them in subsection a – e

Response 7: We revised the opening paragraph to better present the subsections and make it clearer to the readers as below:

Thirty-two studies focused on digital health tools targeting healthcare providers namely physicians as primary intended users [26, 29, 34, 37, 38, 46-53, 57, 58, 61, 63, 64, 68-71, 74, 75, 80, 83, 89, 94, 96, 105, 108, 112] (S3Table). Twenty studies (23/32) explored the effectiveness and implementation of digital health interventions [46-50, 52, 53, 61, 64, 75, 80, 83][57, 58, 63, 68-71] aiming at supporting physicians through decision support algorithm [46, 47], telehealth decision support [37, 48-50, 68-70, 80, 83, 96, 108], mhealth decision support and medical record management [52, 53, 58, 71, 112] or and e-learning [37, 57, 63, 64, 70, 75, 94] while nine studies (9/32) assessed health providers’ readiness and willingness to adopt digital health tools [26, 29, 34, 38, 51, 61, 74, 89, 105][29, 38, 51, 61][26, 34][74]. 

Comment 8: Throughout results, be more specific. For example “The first study (pilot study) noted an overall low reporting and low uptake yet an improvement in BMI and blood pressure reporting; weight, height, blood pressure, and blood sugar measurement; and patient-provider interaction [49].” -what does low reporting and low uptake mean in this context? Sharing more quantitative results will strengthen the paper. And again on page 15 “Overall, the providers were satisfied with e-learning interventions [34, 54, 60] and showed improvement in knowledge [61, 72], yet one study reported challenges related to scheduling and coordination [61].” – could you quantify any of the improvements in knowledge?

Response 8: As we are conducting a scoping review that aim to map out the evidence on digital health and identify knowledge gap, we did not aim to assess the overall effectiveness of the interventions and provide direction of effect. Thus, we did not provide detailed quantitative data. The supplementary files include tables reporting on results of individual papers including detailed quantitative data. However, as the Reviewer suggested to be more specific throughout the results in the manuscript, we added qualitative date where necessary. 

The first study (pilot study) noted an overall low reporting and low uptake yet a significant improvement in BMI and blood pressure reporting in the mHealth application as compared to clinic medical records (47.4% vs 15.8%, p<0.001 and 74.5% vs. 40.7%, p<0.001 respectively); and more frequent reporting of weight, height, blood pressure, and blood sugar measurement; and improved patient-provider interaction [52]. Similarly, for the second study the rate of blood sugar, blood pressure, weight, height and BMI recording improved but not significantly except for BMI (8.1%, P<.001).

Comment 9: It would be nice to have a table that breaks down the topic of the mhealth intervention by target audience.

Response 9: We thank the Reviewer for this constructive comment. As such, we have broken down the topic of mhealth by target audience in the table of characteristics as follow:

Mhealth 34 36%

Clients 21 23%

Providers 7 8%

Clients & providers 3 3%

Data services 2 2%

Health managers 1 1%

Comment 10: “The objective of the intervention was to improve awareness on mental illness among young adults in Afghanistan through educational SMS” – how many of the articles you reviewed did NOT require a smartphone (when the audience was clients)?

Response 10: based on the Reviewer comment, we reviewed the studies involving mobile health and we found that many relied on SMS that do not require the use of smartphones . 

Comment 11: Did any of the articles talk about cell phone or internet limitations, and for articles that were client-facing, the limitations of cell phone ownership in the population? This might be worth addressing in the discussion section. Especially for those who are less familiar with the MENA region, it might be helpful to give the context of whether cell phones are ubiquitous or not, and if not, who doesn’t have them (thus who could not be reached for certain client-facing interventions).

What other implementation issues were there?

Response 11: We reviewed the articles on mhealth and the main limitations of using mhealth reported in these studies are the poor network coverage (connectivity), privacy and illiteracy. We have added these limitations in the results section for studies reporting on mhealth.

“Two observational studies conducted in Lebanon assessed the effectiveness of an intervention including the implementation of disease management guidelines in addition to a mhealth tool for decision support and medical record management in improving the quality of non-communicable disease management for Syrian refugees and vulnerable Lebanese in primary care setting [51, 52]. The first study (pilot study) noted an overall low reporting and low uptake yet an improvement in BMI and blood pressure reporting; weight, height, blood pressure, and blood sugar measurement; and patient-provider interaction [51]. Similarly, for the second study the rate of blood sugar, blood pressure, weight, height and BMI recording improved but not significantly except for BMI. The intervention also resulted in an improvement in the frequency of lifestyle counseling and the quality of patient-provider interaction [52]. The two studies reported the poor connectivity as a main challenge to the use of the application.” Pages 15-16

One study on using text messages suggested coupling voice messages with text messages to assist for illiterate individuals [72]. Page 17

Two observational studies assessed users’ readiness to adopt mhealth tools for maternal and child care in Afghanistan and mental health in Palestine [44, 77]. Results of the study showed that the majority of the respondents owned a mobile phone, were familiar with their services, were open to receiving SMSs and reminders and to call a helpline, and agreed that mhealth can be used to support health [77]. The study found that literacy rate might be a factor influencing mhealth adoption [77]. 

 In the discussion section, we added a sentence reporting on the limitations of mhealth in specific:

Additional challenges include limited computer literacy, unavailability of resources (e.g. staff), weak technological infrastructure and network coverage, restrictive cultural norms and practices, and time burdens. Specific challenges on the use of mhealth included poor connectivity, privacy and illiteracy concerns and illiteracy. These findings concur with another systematic review on the application of m-health technologies in developing countries that stated several limitations, which included interoperability and lack of a technology infrastructure [113].

Regarding the cell phone ownership in the population, two included studies from Palestine and Afghanistan reported on high penetration and access of cell phones among the population. 

Comment 12: Discussion section: page 23 “The implementation of digital health raises concerns such as curtailing privacy of individuals and groups.” – was this addressed throughout the narrative? If not, this isn’t a finding of the scoping review and rather should be placed in a different section of the discussion section – an area that focuses on further considerations, perhaps.

Response 12: Some of the included papers showed a raised concern about privacy among patients and providers. To make sure this sentence in the discussion section better reflect the findings of our scoping review, we have revised the sentence as below:

“Findings of some of the included papers showed a raised concern about privacy among patients and providers”.

 Comment 13: Page 25 – 2nd line – type-o (repeated word)

Response 13: We corrected the typo page 25 – 2nd line.

---

## [Decision Letter · Decision Letter 1]

18 Apr 2023

Digital Health in Fragile States in the Middle East and North Africa (MENA) Region: A Scoping Review of the Literature

PONE-D-22-35597R1

Dear Dr. El-Jardali,

We’re pleased to inform you that your manuscript has been judged scientifically suitable for publication and will be formally accepted for publication once it meets all outstanding technical requirements.

Kind regards,

Sebastien Kenmoe

Academic Editor

PLOS ONE

Additional Editor Comments (optional):

Reviewers' comments:

Reviewer's Responses to Questions

**Comments to the Author**

1. If the authors have adequately addressed your comments raised in a previous round of review and you feel that this manuscript is now acceptable for publication, you may indicate that here to bypass the “Comments to the Author” section, enter your conflict of interest statement in the “Confidential to Editor” section, and submit your "Accept" recommendation.

Reviewer #1: All comments have been addressed

2. Is the manuscript technically sound, and do the data support the conclusions?

Reviewer #1: Yes

3. Has the statistical analysis been performed appropriately and rigorously? 

Reviewer #1: N/A

4. Have the authors made all data underlying the findings in their manuscript fully available?

Reviewer #1: Yes

5. Is the manuscript presented in an intelligible fashion and written in standard English?

Reviewer #1: Yes

6. Review Comments to the Author

Reviewer #1: Thank you for your thorough responses to my questions. I look forward to seeing the paper in publications.

7. PLOS authors have the option to publish the peer review history of their article (what does this mean?). If published, this will include your full peer review and any attached files.

Reviewer #1: No

---

## [Editor Report · Acceptance letter]

20 Apr 2023

PONE-D-22-35597R1 

Digital Health in Fragile States in the Middle East and North Africa (MENA) Region: A Scoping Review of the Literature 

Dear Dr. El-Jardali:

I'm pleased to inform you that your manuscript has been deemed suitable for publication in PLOS ONE. Congratulations! Your manuscript is now with our production department. 

Kind regards, 

on behalf of

Dr. Sebastien Kenmoe 

Academic Editor

PLOS ONE